# Psychometric properties of the brief self-report questionnaire for screening putative pre-psychotic states and validation of clinical utility in young adult

Shih-Kuang Chiang[ID][1]*, Pei-Ti Chen[2], Chen-Chung Liu[3]

1 Department of Counselling and Clinical Psychology, National Dong Hwa University, Hualien County, Taiwan, 2 Department of Psychiatry, Kaohsiung Municipal Kai-Syuan Psychiatric Hospital, Kaohsiung City, Taiwan, 3 Department of Psychiatry, National Taiwan University Hospital and College of Medicine, Taipei City, Taiwan

* skchiang@gms.ndhu.edu.tw

**Data Availability Statement:** All relevant data are within the paper and its Supporting information files.

## Abstract

### Introduction

The Brief Self-Report Questionnaire for Screening Putative Pre-Psychotic States (BQSPS), a brief, self-reported screening tool for risk of psychosis, can detect auditory perceptual disturbances significantly associated with perceived need for psychological services among young adults. However, the relationship is largely explained by the existence of neurotic traits, anxiety and depression symptoms.

### Objective

This study explores possible explanations of previous results from factor structures of the BQSPS and the clinical implications underlying each factor.

### Methods

Construct validity, criterion-related validity, discriminant validity, internal consistency, and test-retest reliability of the BQSPS are determined among young adults ($N = 289$).

### Results

We find that Social Anxiety, Positive Symptoms, and Negative Symptoms are three components in the BQSPS for young adults. Moreover, we find that each component of the BQSPS can be explained by related forms of psychopathology, self-esteem, or personality traits. Finally, the BQSPS can satisfactorily distinguish cases from non-cases using the Symptoms Check List-90-Revised.

### Conclusions

We clarify the clinical implications of each component of the BQSPS and thus expand its clinical utility. The BQSPS has good psychometric properties in young adults from

**Funding:** The author(s) received no specific funding for this work.

**Competing interests:** The authors have declared that no competing interests exist.

an ethnically Chinese population. Limitations and directions for future research are also discussed.

## Introduction

The peak onset for many mental health disorders is young adulthood [1], with the first onset by 25 years of age for 75% of those who will have a mental health disorder [2]. Mental health problems are prevalent among college students, with anxiety disorders being their most common psychiatric problem [3]. Depression is also frequently seen in this population [4], as are various psychotic symptoms [1]. These issues can be related to stressors for college students, including academic load, first-time working, being in a committed personal relationship, and living with others from different cultures and belief systems [5]. Many college students may experience the persistence, increase, or the first onset of mental health and substance use problems [1].

Therefore, developing strategies to identify individuals at high risk of clinical first-episode psychosis is a significant current goal for psychiatric services worldwide [6], especially for young adults. These strategies focus on the early detection of subjects who show only subthreshold symptoms, including positive and negative symptoms or functional difficulties appearing in the prodrome phase of psychosis [7]. Measures for the early detection of people at risk of psychosis have been created and tested during the last two decades [8]. As part of this strategy, the Brief Self-Report Questionnaire for Screening Putative Pre-Psychotic States (BQSPS) was developed [9], with confirmed reliability and validity [8, 10].

The BQSPS targets early and extensive at-risk mental states characterized by subtle symptoms and functional impairments, and it is unlike other questionnaires developed to improve the predictive validity for transition to psychosis [11]. Moreover, in addition to evaluating attenuated positive symptoms, like most screening questionnaires [12], the BQSPS also includes other subthreshold clinical manifestations. Liu et al. suggested two cut-off selections of the BQSPS: (a) respondents checking at least eight items, or (b) those checking three to seven items, including any of three specific items. These two criteria could obtain the largest sensitivity+specificity (0.784+0.705 = 1.489). It has construct validity and can reliably distinguish between clinical and non-clinical samples [9]. Psychometric properties of the BQSPS studies have been verified by two studies. Demmin et al. found moderate to large convergent validity, acceptable internal consistency for each scale, and modest test-retest reliability, recommending its usage for screening psychotic-like experience in college populations [10]. Similarly, Nunez et al. found a stable structure of three correlated factors: social anxiety (SA), negative symptoms (NS), and positive symptoms (PS) in adolescent and young adult subjects [8]. This three-factor model also had the predictive ability for suicidality as an external criterion.

An individual's personality is increasingly recognized as affecting the possibility of psychopathological developments [13]. For example, self-esteem can substantially affect how psychotic symptoms are formed and maintained, as well as recovery from the illness [14]. In particular, low self-esteem may be both a causative factor and a result of a severe mental disorder [15]. This connection is supported by the twin study of Macare et al., which found substantial genetic overlap between schizotypy and neuroticism [16], and the finding of Goodwin et al. that early neuroticism appearing in adolescence may indicate the later development of psychotic symptoms in adulthood [17]. Longitudinal studies have shown both that

neuroticism can increase the risk of psychotic symptoms [18], and conversely that extraversion can help to avoid or mitigate depression and social anxiety [19].

The Symptom Checklist-90-R (SCL-90-R) is a self-rating-scale for assessing general psychopathology and specific symptoms [20], used to distinguish potential psychosis-like pathology as a short screening tool for pre-psychotic states [21]. It has been used as a valid indicator of prodromal episodes [22] and successfully detected a disposition to psychosis [23], by now being a standard measure for susceptibility to psychosis [24].

Since there is a recognized need for developing shorter questionnaires with robust psychometric properties [12, 25] the current study uses exploratory factor analyses to examine the internal structure of the BQSPS and compares the results with Nunez et al. [8]. We then test associations between the BQSPS and related criteria to determine their contributions to the explained variance of the BQSPS. In addition, we use receiver operating characteristic (ROC) analysis to investigate the discrimination of the BQSPS in a young adult population and compare the results with Müller et al. [21]. Finally, we also examine the internal constancy and test-retest reliability of the BQSPS and compare them with Demmin et al. [10].

## Materials and methods

### Procedure

The Institutional Review Board of Yuli Hospital approved this study (Approval number: YLH-IRB-10502). We had obtained permission from the original copyright holder of the BQSPS before this study began. All participants ($N$ = 289) completed an informed consent procedure and the BQSPS. A subsample ($N$ = 219) also completed the Rosenberg Self-Esteem Scale, Eysenck Personality Questionnaire, and the SCL-90-R at this time. A subsample of the first subsample ($N$ = 70) completed the BQSPS again, two weeks after the first administration.

### Participants

We recruited the subjects by stratified random sampling. There were 300 undergraduate students from a representative university in eastern Taiwan who participated in this study. The students represented a wide range of faculties, including Chinese, English, Clinical and Counseling Psychology, Chemistry, Life Science, Electrical Engineering, Computer Science & Information Engineering, Business Administration, Finance, Accounting, Tourism Recreation & Leisure Study, Educational Administration and Management, Special Education, Physical Education and Kinesiology, Music, Art & Design. We eliminated 11 responses due to incomplete answers, leaving 289 participants who participated in the following analysis. There were 107 men (37%) and 182 women, whose ages ranged from 19 to 23 years, with a mean of 20.65 (SD = 0.89). Their education ranged from 13 to 17 years, with a mean of 14.65 (SD = 0.89). We used data from the sample to explore the construct validity and internal consistency of the BQSPS. A subsample by simple random sampling ($N$ = 70) was used to assess the test-retest reliability of the BQSPS. The remaining subjects ($N$ = 219) were used to test the criterion-related validity and discriminant validity of the BQSPS.

### Measures

**Rosenberg Self-Esteem Scale (RSES).** Rosenberg developed the original Rosenberg Self-Esteem Scale (RSES) [26], which contains ten items, each with a 4-point scale from 1, strongly disagree, to 4, strongly agree. A higher score means that the subject has higher self-esteem. The Chinese version of the RESR also showed good Cronbach's α (= .85) [27] and construct

validity [28]. In the current study, we used the Chinese version of the RSES to explore the criterion-related validity of the BQSPS by measuring participants' self-esteem.

**Eysenck Personality Questionnaire (EPQ).** Eysenck and Eysenck developed the original Eysenck Personality Questionnaire (EPQ) [29]. The original EQS contains five subscales, including psychoticism, extraversion, neuroticism, and lying, a total of 90 items. Lu (1995) developed a Chinese short-form version of the EQS [30] with good Cronbach's α (= .90) and construct validity [31]. It contains 25 items that pertain to either neuroticism or extraversion factors. In this study, the Chinese short-form version of the EQS was used to explore the criterion-related validity of the BQSPS by measuring participants' neuroticism and extraversion.

**The Symptom Checklist-90-R (SCL-90-R).** The English version of the Symptom Checklist-90 (SCL-90) was created by Derogatis [20] and was revised to SCL-90-R [32]. The original SCL-90-R contains 90 items, and each item has a 5-point scale from 0, which means no symptoms, to 4, which means strong symptoms. The original SCL-90-R had good psychometric properties [33, 34]. Yeh's Chinese version SCL-90-R also had good psychometric characteristics and norms [35]. Cronbach's α and test-retest reliability of nine symptom dimensions ranged from.77 to.90 and.70 to.93. The original SCL-90-R included nine symptom dimensions: somatization, obsessive-compulsion, interpersonal sensitivity, depression, anxiety, hostility, phobic anxiety, paranoid ideation, and psychoticism. These symptoms could be calculated into three indexes: the Global Severity Index (GSI), the Positive Symptom Distress Index (PSDI), and the Positive Symptom Total (PST). A T score for the GSI above 63 points, or the T score of any two symptoms dimension above 63 points, generally indicates a significant clinical psychological problem [36]. The current study uses the Chinese version of the SCL-90-R to explore the criterion-related validity of the BQSPS by measuring participants' symptoms and severity.

**The Putative Pre-Psychotic State Scale (BQSPS).** Liu et al. developed the Putative Pre-Psychotic State Scale (BQSPS) [9]. The BQSPS contains 15 items, with each item answered using "yes" or "no" to minimize the response burden. A "yes" indicates an affirmative response to the item of a deviant experience. In this study, we used the BQSPS to analyze its construct validity, criterion-related validity, discriminant validity, internal consistency, and test-retest reliability.

## Statistical analyses

Independent sample t-test analysis, exploratory factor analysis, Pearson correlation analysis, stepwise regression analysis, internal consistency analysis, test-retest reliability analysis, and ROC analysis used the Statistical Package for the Social Sciences (SPSS), version 14.

## Results

Because the test sample was found to have good sampling properties (*KMO* = .80; Bartlett's test, *p* < .001), we adopted principal component analysis with an oblique factor rotation. The scree plot and eigenvalues from the initial factor extraction indicated that a three-component solution explained 45.51% of the variance. According to Tabachnick & Fidell, using an alpha level of.01 (two-tailed), a rotated factor loading for a sample size of at least 300 would need to be at least.32 to be considered statistically meaningful [37]. In this study, we retain an item base on its factor loading at least.40.

Table 1 shows that item 7 had loading (< .40) on component 1. But item 7 had the highest loading (.40) on component 2. After adding item 7 into component 2, Cronbach's alpha improved from.59 to.60. Therefore, we attributed item 7 to component 2. Finally, component 1 has six items, component 2 has five items, and component 3 has four items. The first

**Table 1. Exploratory factor analysis of the putative pre-psychotic states scale.**

| Item number and abbreviated wording | Component[a] | | |
|---|---|---|---|
| | 1 | 2 | 3 |
| 12. I am poor at returning social courtesies and gestures. (SA) | .72[b] | -.14 | -.04 |
| 5. I am mostly quiet when with others. (SA) | .69[b] | -.03 | .01 |
| 1. I cannot deal with the pressures associated with crowds. (SA) | .64[b] | .12 | .00 |
| 11. I do not have an expressive and lively way of speaking. (SA) | .60[b] | .08 | .08 |
| 8. I feel nervous when giving a speech in front of a large group of people. (SA) | .58[b] | -.02 | -.08 |
| 2. I feel I cannot get close to people. (SA) | .48[b] | .03 | -.39 |
| 14. Do you often pick up hidden threats or put downs from the words or actions of others? (PS) | .12 | .72[b] | -.04 |
| 6. I sometimes become concerned about the loyalty and trustworthiness of friends or coworkers. (PS) | .02 | .70[b] | -.06 |
| 13. When you see people talking to each other, do you often wonder if they are talking about you? (PS) | .06 | .62[b] | .01 |
| 15. Do you hear some sounds, voices, or calls of your name when nobody is around you? (PS) | -.21 | .55[b] | -.01 |
| 7. I tend to keep my feelings to myself. (SA) | .33 | .40[b] | .01 |
| 4. I feel mentally insufficient and easily fatigued while thinking or reading. (NS) | -.14 | -.00 | -.80[b] |
| 9. I cannot focus on a task and need to take frequent breaks while working (studying). (NS) | -.05 | -.01 | -.70[b] |
| 3. I feel lethargic whatever I do. (NS) | .05 | .06 | -.70[b] |
| 10. I always mess up whatever I do. (NS) | .18 | .03 | -.62[b] |
| Variance explained (%): | 22.98 | 10.68 | 9.85 |

*Note*. SA = Social Anxiety; PS = Positive Symptoms; NS = Negative Symptoms.
[a]Component 1 = Social Anxiety; Component 2 = Positive Symptoms; Component 3 = Negative Symptoms
[b]Item has a high loading on the corresponding component.

component is composed of six items from the Social Anxiety Scale of Nüñez et al. [8]. The item loadings range from.48 to.72, while the component explains 24.98% of the total variance. The second component is composed of four items from the Positive Symptoms Scale, with one item from the Social Anxiety Scale of Nüñez et al. The range of item loadings is from.40 to.72, and the component explains 10.68% of the total variance. Four items of the Negative Symptoms Scale from Nüñez et al. had high loading on the third component. The item loadings ranged from.62 to.80, with the component explaining 9.85% of the total variance. The components structure and contents of each component are very similar to Nüñez et al.

Table 2 shows the Pearson correlations between scores on the three BQSPS components and scores on the two other instruments (the RSES and the EPQ), with the SCL-90-R in the first sample subgroup (*N* = 219). As predicted, scores on three components and the total scale of the BQSPS were negatively and significantly associated with scores on the RESE and Eysenck-Extraversion Subscale, confirming the criterion-related validity of the BQSPS. Also consistent with predictions, scores on three components and the total scale of the BQSPS were positively and strongly associated with scores on the Eysenck-Neuroticism Subscale. Table 3 shows that scores on three components and the total scale of the BQSPS were positively and significantly associated with scores on all the symptom dimensions of the SCL-90-R.

In the current study, we executed a stepwise regression analysis in two stages. First, we predicted three components of the BQSPS by using symptoms dimensions of the SCL-90-R as predictors. We further added self-esteem, extraversion, neuroticism, and predicted three

**Table 2. Pearson correlations between scores on the putative pre-psychotic states scale and other psychological measures, and SCL-90-R (subgroup of first sample).**

| External measures | Social Anxiety | Positive Symptoms | Negative Symptoms | Total |
|---|---|---|---|---|
| RSES | -.47*** | -.39*** | -.52*** | -.61*** |
| Eysenck-Extraversion Subscale | -.71*** | -.17* | -.20** | -.52*** |
| Eysenck-Neuroticism Subscale | .44*** | .53*** | .55*** | .66*** |
| SCL-90-R | | | | |
| Somatization | .23** | .39*** | .37*** | .43*** |
| Obsessive-Compulsive | .39*** | .46*** | .63*** | .63*** |
| Interpersonal Sensitivity | .44*** | .60*** | .58*** | .70*** |
| Depression | .38*** | .51*** | .66*** | .66*** |
| Anxiety | .35*** | .46*** | .54*** | .58*** |
| Hostility | .21** | .51*** | .49*** | .51*** |
| Phobic Anxiety | .34*** | .46*** | .45*** | .54*** |
| Paranoid Ideation | .29*** | .63*** | .43*** | .57*** |
| Psychoticism | .32*** | .56*** | .50*** | .59*** |

*$p < .05$,
**$p < .01$,
***$p < .001$

components of the BQSPS again. Table 3 shows that variance was explained for the two models of each component of the BQSPS and their beta values.

First, we use a ROC analysis to test the utility of criterion (a). The results show that the best cut-off score is above 7.5 scores with .87 of the area under the curve (AUC) (95% CI:.80-.93) ($p < .001$). The sensitivity and specificity are .80 and 81, respectively. We concurrently compared

**Table 3. Stepwise regression analysis of components of the putative pre-psychotic states scale.**

| Predictor | Social Anxiety | | Positive Symptoms | | Negative Symptoms | |
|---|---|---|---|---|---|---|
| | Model 1 | Model 2 | Model 1 | Model 2 | Model 1 | Model 2 |
| | Beta | Beta | Beta | Beta | Beta | Beta |
| SCL-90-R | | | | | | |
| Somatization | -.02 | .03 | -.03 | .01 | -.14 | -.08 |
| Obsessive-Compulsive | .23* | .08 | -.15 | -.27** | .30** | .23* |
| Interpersonal Sensitivity | .46*** | .29*** | .30*** | .32** | .16 | .08 |
| Depression | .14 | .07 | -.11 | -.10 | .54*** | .42*** |
| Anxiety | .05 | .04 | -.15 | -.15 | -.06 | -.08 |
| Hostility | -.29** | -.15* | .05 | .08 | -.05 | -.04 |
| Phobic Anxiety | .10 | .01 | -.03 | -.08 | .01 | -.03 |
| Paranoid Ideation | -.07 | -.02 | .40*** | .42*** | -.18* | -.18* |
| Psychoticism | -.06 | .04 | .003 | .07 | -.07 | -.05 |
| Self-Esteem | | -.09 | | -.07 | | -.20** |
| Extraversion | | -.60*** | | -.02 | | .02 |
| Neuroticism | | .21*** | | .30*** | | .14* |
| $R^2$ | .23 | .61 | .43 | .49 | .46 | .52 |

*$p < .05$,
**$p < .01$,
***$p < .001$

**Table 4. Results of ROC analysis of BQSPS and subscales.**

| Scale | AUC(95% CI) | Best cutoff | Sensitivity | Specificity |
|-------|-------------|-------------|-------------|-------------|
| SA | .72 (.63, .82) | ≥ 3.5 | .55 | .827 |
| PS | .79 (.71, .87) | ≥ 1.5 | .825 | .659 |
| NS | .82 (.74, .89) | ≥ 2.5 | .75 | .765 |
| BQSPS | .87 (.80, .93) | ≥ 7.5 | .80 | .81 |

the results of the ROC analysis of SA, PS, NS with BQSPS (see Table 4 and Fig 1). The results support that criterion (a) is appropriate for the sample in the current study. Second, we compare the utility of criterion (a), criterion (b), and the combination of these two by Chi-Square analysis. Table 5 shows that using criterion (a) has the largest value of sensitivity + specificity. It is noteworthy that using the combined criteria could yield the highest sensitivity to the most significant false-positive and the fewest false negatives.

As shown in Table 6, Cronbach's alpha for the BQSPS and its components ranged from .60 to .78 for the main sample in the study. This suggests that internal consistency for the BQSPS varies from acceptable to good. Table 6 also shows the 2-week test-retest reliability, whose values ranged from $r = .67$ to $r = .82$.

## Discussion

In the current study, we found that there were three components in our sample. The contents of each component were almost the same as the previous study [8], except that item 7 was moved from social anxiety to positive symptoms. However, because item 7 had a high positive correlation with PS and paranoid ideation, neuroticism and interpersonal sensitivity had larger impacts on PS. Item 7 might reflect a person's behavioural results derived from his paranoid ideation, neuroticism, or interpersonal sensitivity. This verifies that a stable three-component structure of the BQSPS exists in a normal young population across language. We followed Cohen's suggestion to interpret a correlation of .1 as small, .3 as moderate, and .5 as large [38]. As expected, relationships with convergent scales, including neuroticism and all symptoms dimensions, ranged from small to large. Also, relationships with discriminant scales, including self-esteem and extroversion, ranged from small to large. This demonstrates that the BQSPS has good criterion-related validities that fit the theoretical psychological expectations. The results are comparable to a previous study using established psychosis screens and un-associated questionnaires to determine construct validity of the BQSPS [10].

We also found some selected criteria contributing to the explained variance on SA, PS, NS. First, extraversion, interpersonal sensitivity, neuroticism, and hostility had significant impacts on SA. Together, they explain 61% of the variance of SA. We note that personality dimensions had significantly larger impacts than symptom dimensions on SA. We found that extraversion was protective against social anxiety, as in a previous study [19]. Second, paranoid ideation, interpersonal sensitivity, neuroticism, and obsessive-compulsion had significant impacts on PS. Together, they can explain the 49% variance of PS. Previous review studies consistently found that paranoia is associated with more negative conscious self-concepts [39, 40]. Our data also supported that finding ($r = -.38$, $p < .001$). Third, depression, obsessive-compulsion, self-esteem, paranoid ideation, and neuroticism had significant impacts on NS. Together, they can explain 52% of the variance of NS. NS addresses four items referring to feelings of tiredness, lethargy and concentration difficulty. These descriptions are similar to some of the symptoms of a depression episode. Our findings showed that there was a strong association between

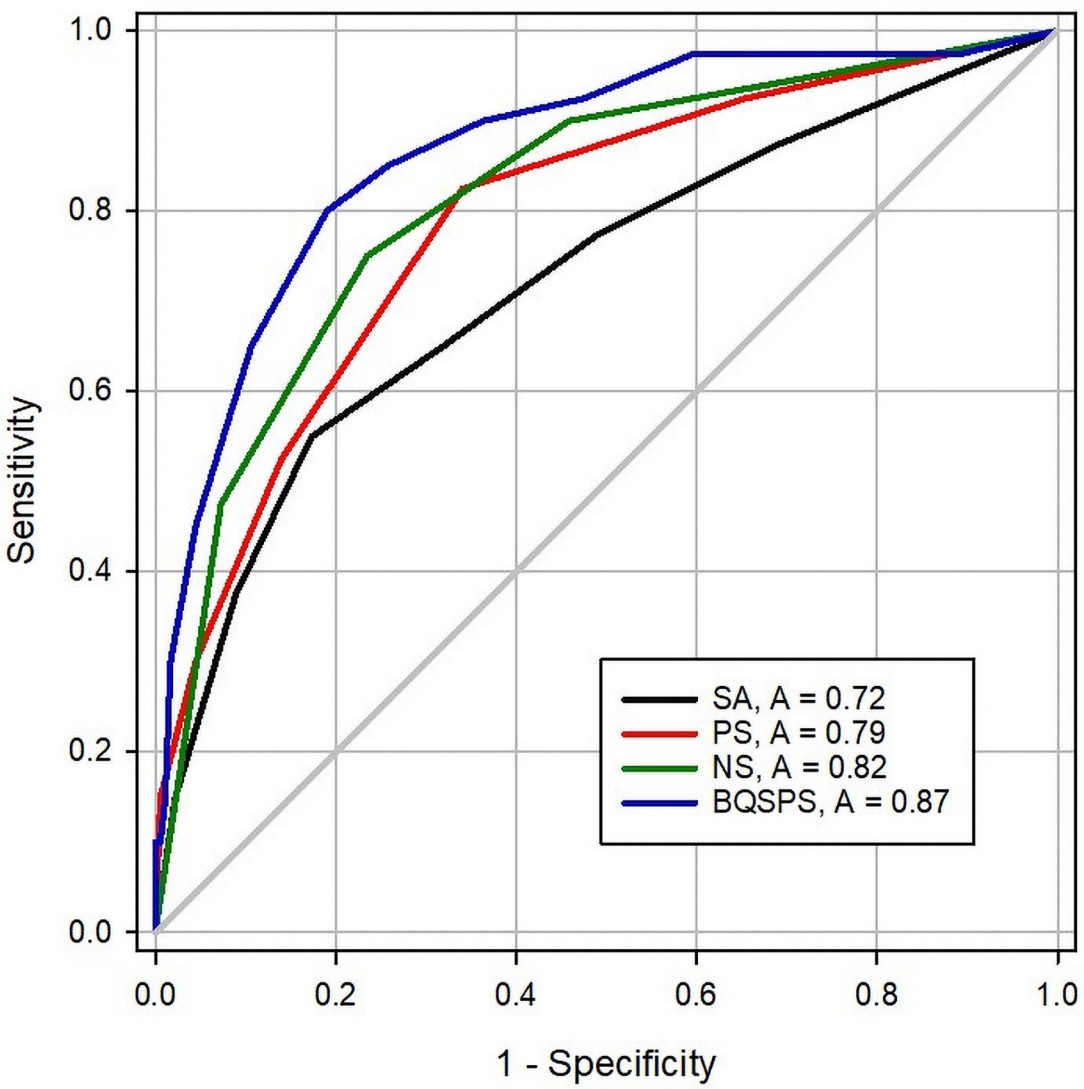

**Fig 1. ROC curves of SA, PS, NS, and BQSPS.**

depression and NS. Finally, we found that neuroticism had a pervasive impact on SA, PS, NS. In summary, the BQSPS can reflect some important predictors for psychopathology.

We used the SCL-90-R to differentiate our sample into the two groups of psychological distress and no-distress. ROC analysis showed that SA (.72), PS (.79), NS (.82), and BQSPS (.87) had acceptable to excellent discriminant validities. The BQSPS had the highest AUC by using the cut-off in Liu et al. [9]. We also found that combining two screening criteria in Liu et al.'s study could increase the sensitivity of the BQSPS to .95. We compared the results with Müller et al. [21]. Müller et al. developed a new screening tool (Self-screen-Prodrome, SPro) to differentiate cases and non-cases, using the SCL-90-R-subscales of psychoticism [PSYC] and

**Table 5. Chi-square analysis of the putative pre-psychotic states scale and SCL-90-R.**

| | | Criterion (a) | | Criterion (b) | | Combined (a)&(b) | | |
|---|---|---|---|---|---|---|---|---|
| | | **No** | **Yes** | **No** | **Yes** | **No** | **Yes** | **Total** |
| SCL-90-R | No | 145 | 34 | 108 | 71 | 104 | 75 | 179 |
| | Yes | 8 | 32 | 5 | 35 | 2 | 38 | 40 |
| | *Chi-square* | 57.79*** | | 29.96*** | | 36.91*** | | |
| Sensitivity | | .80 | | .875 | | .95 | | |
| Specificity | | .81 | | .60 | | .58 | | |
| False Negative | | .20 | | .125 | | .05 | | |
| False Positive | | .19 | | .40 | | .42 | | |

***$p < .001$

**Table 6. Internal consistency and test-retest reliability of the putative pre-psychotic states scale and its components.**

| | Internal consistency (α) | Test-retest reliability (r) |
|---|---|---|
| | 1$^{st}$ Sample ($n = 279$) | Subgroup of 1$^{st}$ Sample ($n = 70$) |
| Social Anxiety | .73 | .71*** |
| Positive Symptoms | .60 | .67*** |
| Negative Symptoms | .69 | .76*** |
| Total | .78 | .82*** |

***$p < .001$

paranoid ideation [PARA]≥63 as criteria [21]. They found that the SPro subscale for psychotic risk (SPro-Psy-Risk) could identify cases best, with a sensitivity of 74% and a specificity of 61%. Apparently, the sensitivity of the BQSPS is better than SPro-Psy-Risk's for identifying cases. We thought this is due to criteria to define cases in the current study are more comprehensive than those of Müller et al. These criteria reflected appropriately aims of the BQSPS, which try to detect early and broadly at-risk mental states characterized by subtle symptoms and functional impairments [9]. The BQSPS had acceptable to good internal consistency and good test-retest reliability, similar to findings of Müller et al. [21].

However, the study has two limitations. First, our subjects were undergraduate college students, and whether the findings could be generalized to other young populations remains to be tested. Second, since the BQSPS did not originally focus on increasing the predictive validity of transition from at-high-risk to psychosis, a longitudinal study is needed to confirm the predictive validity of the BQSPS for the onset of psychosis.

## Conclusion

To conclude, this study shows that the BQSPS has good psychometric properties for young adults. We also clarify clinical implications for each component of BQSPS and thereby expand its clinical utility in university settings.

## Supporting information

**S1 Dataset. Anonymized dataset.**
(XLS)

## Author Contributions

**Conceptualization:** Shih-Kuang Chiang, Chen-Chung Liu.

**Formal analysis:** Pei-Ti Chen.

**Investigation:** Pei-Ti Chen, Chen-Chung Liu.

**Methodology:** Shih-Kuang Chiang, Chen-Chung Liu.

**Resources:** Chen-Chung Liu.

**Software:** Pei-Ti Chen.

**Supervision:** Shih-Kuang Chiang.

**Writing – original draft:** Shih-Kuang Chiang.

**Writing – review & editing:** Shih-Kuang Chiang.

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
