## [Decision Letter · Decision Letter 0]

13 Apr 2021

PONE-D-20-34283

Psychometric properties of the brief self-report questionnaire for screening putative pre-psychotic states and validation of clinical utility in young adult

PLOS ONE

Dear Dr. Chiang,

Thank you for submitting your manuscript to PLOS ONE. After careful consideration, we feel that it has merit; indeed, both the Reviewers and myself have considered your study of interest for its clinical implications. However, it does not fully meet PLOS ONE’s publication criteria as it currently stands. Therefore, we invite you to submit a revised version of the manuscript that addresses the points raised during the review process.

We look forward to receiving your revised manuscript.

Kind regards,

Paola Gremigni, Ph.D.

Academic Editor

PLOS ONE

Journal Requirements:

2. We note that Table 1 may include questionnaire items that may have been previously published. The reproduction of previously published work has implications for the copyright that may apply to these publications. We would be grateful if you could clarify whether you have obtained permission from the original copyright holder to republish these items under a CC BY license. If you have not obtained permission to publish these items please remove them from your manuscript. You may wish to replace the text you have removed with relevant question numbers/ brief descriptions of each item; please be sure to include any relevant references and in-text citations.

Additional Editor Comments:

1) Please, use "construct validity" instead of "constructive validity".

2) Please, use "components" instead of "factors" as you run PCA not EFA.

3) Please, do not say that factor loadings were significant, as you did not report their p values. Usually, we retain an item based on a pre-established cut-point (e.g. >.30 or >. 40) not based on statistical significance.

4) In the Discussion, a mention of your study limitations is missing; therefore, please, add this part.

Reviewers' comments:

Reviewer's Responses to Questions

**Comments to the Author**

1. Is the manuscript technically sound, and do the data support the conclusions?

Reviewer #1: Yes

Reviewer #2: Yes

2. Has the statistical analysis been performed appropriately and rigorously? 

Reviewer #1: Yes

Reviewer #2: Yes

3. Have the authors made all data underlying the findings in their manuscript fully available?

Reviewer #1: Yes

Reviewer #2: Yes

4. Is the manuscript presented in an intelligible fashion and written in standard English?

Reviewer #1: Yes

Reviewer #2: No

5. Review Comments to the Author

Reviewer #1: Identification of psychosis in the early stage is very much fruitful regarding the clinical outcome. So, this study is a good effort to meet the burning issue.

The study was found well organized, used various appropriate statistical analysis tools, and was well written.

I would like to provide some minor suggestions-

1)The full form of ROC should be mentioned at the beginning

2)Psychometric properties of all Chinese versions should be specifically mentioned (Cronbach’s alpha etc.)

3)The statistical part seemed perfect from my point of view, but require further scrutiny by other experts.

Reviewer #2: I understand that this study was performed to validate the psychometric properties of the BQSPS by comparing the results with 2 other previous studies which had the same objective yet a distinctively different population..

The main objective of the study was feasible and interesting yet in the execution I found myself to be lost ..

As the author compared some other studies which used different scales and although the comparisons might be richer this way ..the main objective is lost in the way and the discussion section became overcrowded and confusing.. focusing on the author’s main objective can make the reader more interested in the results.

Table 1 is not well organized, some points are affirmatives others are questions.

Table 2& 3 are better to follow the same items sequence not to have the reader confused .

Finally, the manuscript should be reviewed by a native English speaker for the quality of language.

6. PLOS authors have the option to publish the peer review history of their article (what does this mean?). If published, this will include your full peer review and any attached files.

Reviewer #1: **Yes: **Panchanan Acharjee

Reviewer #2: No

---

## [Author Response · Author response to Decision Letter 0]

28 Apr 2021

To academic editor

Answer: We had checked and confirmed that our manuscript meets PLOS ONE’s style requirements. 

2. We note that Table 1 may include questionnaire items that may have been previously published. The reproduction of previously published work has implications for the copyright that may apply to these publications. We would be grateful if you could clarify whether you have obtained permission from the original copyright holder to republish these items under a CC BY license. If you have not obtained permission to publish these items please remove them from your manuscript. You may wish to replace the text you have removed with relevant question numbers/ brief descriptions of each item; please be sure to include any relevant references and in-text citations.

Answer: The third author of this manuscript is the original copyright holder of the BQSPS. Without any doubt, we got his permission before this study began. We also clarify this in the manuscript.

Answer: We do not change our Data Availability statement. However, we changed our data sharing method from providing a DOI number into providing Supporting information files in this study.

Answer: We did not cite any articles that have been retracted. For answering the third point of additional editor comments, we add a new citation, “Tabachnick BG, Fidell LS. Using multivariate statistics (5th ed.). Boston, MA: Allyn & Bacon; 2007”.

Additional Editor Comments:

1) Please, use "construct validity" instead of "constructive validity".

2) Please, use "components" instead of "factors" as you run PCA not EFA.

3) Please, do not say that factor loadings were significant, as you did not report their p values. Usually, we retain an item based on a pre-established cut-point (e.g. >.30 or >. 40) not based on statistical significance.

4) In the Discussion, a mention of your study limitations is missing; therefore, please, add this part.

Answer: We have revised these minor issues in the manuscript accordingly.

To reviewer 1

1. Is the manuscript technically sound, and do the data support the conclusions?

Reviewer #1:Yes

Answer: Thank the reviewer's comment.

2. Has the statistical analysis been performed appropriately and rigorously?

Reviewer #1:Yes

Answer: Thank the reviewer's comment.

3. Have the authors made all data underlying the findings in their manuscript fully available?

Reviewer #1:Yes

Answer: Thank the reviewer's comment.

4. Is the manuscript presented in an intelligible fashion and written in standard English?

Reviewer #1:Yes

Answer: Thank the reviewer's comment.

5. Review Comments to the Author

Reviewer #1: Identification of psychosis in the early stage is very much fruitful regarding the clinical outcome. So, this study is a good effort to meet the burning issue.

The study was found well organized, used various appropriate statistical analysis tools, and was well written.

I would like to provide some minor suggestions-

1)The full form of ROC should be mentioned at the beginning

2)Psychometric properties of all Chinese versions should be specifically mentioned (Cronbach’s alpha etc.)

3)The statistical part seemed perfect from my point of view, but require further scrutiny by other experts.

Answer: Thank the reviewer's comment. For suggestions #1 and #2, we have revised these minor issues in the manuscript accordingly. We also expected to get a critical suggestion about suggestion 3.

6. PLOS authors have the option to publish the peer review history of their article (what does this mean?). If published, this will include your full peer review and any attached files.

Do you want your identity to be public for this peer review? For information about this choice, including consent withdrawal, please see our Privacy Policy.

Reviewer #1:Yes: Panchanan Acharjee

Answer: Thank the reviewer's comment.

To reviewer 2

1. Is the manuscript technically sound, and do the data support the conclusions?

Reviewer #2:Yes

Answer: Thank the reviewer's comment.

2. Has the statistical analysis been performed appropriately and rigorously?

Reviewer #2:Yes

Answer: Thank the reviewer's comment.

3. Have the authors made all data underlying the findings in their manuscript fully available?

Reviewer #2:Yes

Answer: Thank the reviewer's comment.

4. Is the manuscript presented in an intelligible fashion and written in standard English?

Reviewer #2:No

Answer: Thank the reviewer's comment. An experienced academic native English speaker edited the paper.

5. Review Comments to the Author

Reviewer #2: I understand that this study was performed to validate the psychometric properties of the BQSPS by comparing the results with 2 other previous studies which had the same objective yet a distinctively different population..

The main objective of the study was feasible and interesting yet in the execution I found myself to be lost ..

As the author compared some other studies which used different scales and although the comparisons might be richer this way ..the main objective is lost in the way and the discussion section became overcrowded and confusing.. focusing on the author’s main objective can make the reader more interested in the results.

Table 1 is not well organized, some points are affirmatives others are questions.

Table 2& 3 are better to follow the same items sequence not to have the reader confused .

Finally, the manuscript should be reviewed by a native English speaker for the quality of language.

Answer: Thank the reviewer's comment. We had added some sentences to make the paragraph more clear for the readers. Besides, an experienced academic native English speaker edited the paper.

6. PLOS authors have the option to publish the peer review history of their article (what does this mean?). If published, this will include your full peer review and any attached files.

Do you want your identity to be public for this peer review? For information about this choice, including consent withdrawal, please see our Privacy Policy.

Reviewer #2:No

Answer: Thank the reviewer's comment.

---

## [Editor Report · Decision Letter 1]

6 May 2021

Psychometric properties of the brief self-report questionnaire for screening putative pre-psychotic states and validation of clinical utility in young adult

PONE-D-20-34283R1

Dear Dr. Chiang,

We’re pleased to inform you that your manuscript has been judged scientifically suitable for publication and will be formally accepted for publication once it meets all outstanding technical requirements.

Kind regards,

Paola Gremigni, Ph.D.

Academic Editor

PLOS ONE

Additional Editor Comments (optional):

The Authors answered adequately the minor concerns expressed by the reviewers and the academic editor. The statistical approach is appropriate.

---

## [Editor Report · Acceptance letter]

8 Jun 2021

PONE-D-20-34283R1 

Psychometric properties of the brief self-report questionnaire for screening putative pre-psychotic states and validation of clinical utility in young adult 

Dear Dr. Chiang:

I'm pleased to inform you that your manuscript has been deemed suitable for publication in PLOS ONE. Congratulations! Your manuscript is now with our production department. 

Kind regards, 

on behalf of

Prof. Paola Gremigni 

Academic Editor

PLOS ONE